# Fast and Automatic Reconstruction of Semantically Rich 3D Indoor Maps from Low-quality RGB-D Sequences

**DOI:** 10.3390/s19030533

**Published:** 2019-01-27

**Authors:** Shengjun Tang, Yunjie Zhang, You Li, Zhilu Yuan, Yankun Wang, Xiang Zhang, Xiaoming Li, Yeting Zhang, Renzhong Guo, Weixi Wang

**Affiliations:** 1Research Institute for Smart Cities & Shenzhen Key Laboratory of Spatial Information Smart Sensing and Services, School of Architecture and Urban Planning, Shenzhen University, Shenzhen 518060, China; shengjuntang@szu.edu.cn (S.T.); RISC@szu.edn.cn (Y.Z.); boycecug@gmail.com (Y.L.); yuanzl13@szu.edu.cn (Z.Y.); yankunwang@szu.edu.cn (Y.W.); lixming@szu.edu.cn (X.L.); guorz@szu.edu.cn (R.G.); 2State Key Laboratory of Information Engineering in Surveying Mapping and Remote Sensing, Wuhan University, Wuhan 430000, China; zhangxiangsw@whu.edu.cn (X.Z.); zhangyeting@263.net (Y.Z.); 3Key Laboratory of Urban Land Resources Monitoring and Simulation, Ministry of Land and Resources, Shenzhen 518040, China

**Keywords:** mobile mapping, indoor reconstruction, geometric computation, point cloud, space partition

## Abstract

Semantically rich indoor models are increasingly used throughout a facility’s life cycle for different applications. With the decreasing price of 3D sensors, it is convenient to acquire point cloud data from consumer-level scanners. However, most existing methods in 3D indoor reconstruction from point clouds involve a tedious manual or interactive process due to line-of-sight occlusions and complex space structures. Using the multiple types of data obtained by RGB-D devices, this paper proposes a fast and automatic method for reconstructing semantically rich indoor 3D building models from low-quality RGB-D sequences. Our method is capable of identifying and modelling the main structural components of indoor environments such as space, wall, floor, ceilings, windows, and doors from the RGB-D datasets. The method includes space division and extraction, opening extraction, and global optimization. For space division and extraction, rather than distinguishing room spaces based on the detected wall planes, we interactively define the start-stop position for each functional space (e.g., room, corridor, kitchen) during scanning. Then, an interior elements filtering algorithm is proposed for wall component extraction and a boundary generation algorithm is used for space layout determination. For opening extraction, we propose a new noise robustness method based on the properties of convex hull, octrees structure, Euclidean clusters and the camera trajectory for opening generation, which is inapplicable to the data collected in the indoor environments due to inevitable occlusion. A global optimization approach for planes is designed to eliminate the inconsistency of planes sharing the same global plane, and maintain plausible connectivity between the walls and the relationships between the walls and openings. The final model is stored according to the CityGML3.0 standard. Our approach allows for the robust generation of semantically rich 3D indoor models and has strong applicability and reconstruction power for complex real-world datasets.

## 1. Introduction

In recent years, semantically rich digital 3D indoor models have been increasingly used for indoor mapping and navigation, building management, simulation and virtual reality [1,2,3]. The creation of a 3D indoor model involves measuring the geometric attributes of the original scanning data and transforming those measurements into a geometrically consistent and semantically rich representation [4]. Laser scanners are widely used for distance measurements of surfaces visible from the sensor’s viewpoint. To obtain sufficient coverage of the spaces, a terrestrial laser scanner is placed in different locations throughout the facility and the point clouds from each station are merged into one via geometric registration. Consumer-level 3D acquisition devices such as mobile laser scanners and RGB-D sensors are now available for fast, accurate data collection [5,6,7,8], and greatly reduce the time-cost of data collection [9,10]. 

However, reconstruction of complex indoor scenes poses great challenges, and a method of automatic indoor modeling from point clouds is urgently needed. The main area of research interest is the problem of determining the architectural structure of indoor scenes (e.g., room spaces, walls, floors, ceilings, doors, and windows). Some applications such as path panning and indoor navigation require detailed relationships between build elements, such as wall connectivity or containing relationships between doors and walls. The process of converting point cloud data into a semantically rich 3D indoor model requires extensive use of computer vision techniques such as geometric segmentation and clustering, element labeling, and spatial relationship recovery [11,12,13]. Due to the complex structure layouts, clutter and occlusions, indoor reconstruction presents a numbers of distinctive challenges that make it significantly harder to manage than outdoor reconstruction [10,14]. First, to handle such occluded areas, 3D scanners collect huge point clouds from close distances and multiple directions, and these often cause system slow-down or failure [15]. Second, the indoor reconstruction method must be highly tolerant of missing data from occlusions and clutter during data collection. This makes it hard to obtain the information about walls, floors, and other structures of interest. Furthermore, there is a challenge of recovering the interior structure and topology in terms of room space, connectivity between rooms, and the containing or adjacency relationships of different indoor components. 

Some research on indoor environments focuses on the classification and labeling of the objects with deep learning methods [16,17], while the reconstruction of building components has been neglected. The conventional 3D modeling process requires modelers to subjectively determine building object shapes through manual creation, which leads to low productivity and erroneous modeling. To cope with unstructured and even incomplete point clouds, most reconstruction methods make use of “PreDefined” geometric knowledge of the interior entities to facilitate walls, floors, and ceiling recognition and reconstruction [1,5,18]. However, many existing solutions are targeted at single rooms or simply connected environments and cannot automatically reconstruct room spaces and their relationships in more complex environments [19,20]. 

This paper makes full use of information from RGB-D sequences containing RGB-D frames, colorized point clouds, and camera trajectories to reconstruct a semantically rich 3D indoor model. Our method is capable of identifying and modeling the main structural components of complex indoor environments such as spaces, walls, floors, ceilings, windows, and doors from RGB-D datasets. A review of the literature on 3D indoor reconstruction methods based on point clouds is presented in Section 2. Section 3 describes how indoor elements can be automatically derived from observation data with an extremely high level of clutter. In Section 4, the applicability and reconstruction power of the proposed procedures is demonstrated on complex real-world datasets. Finally, in the concluding Section 5, the strengths and limitations of the proposed method are discussed. 

## 2. Related Work

Because they are fast, simple to use, and highly accurate, mobile 3D scanners are widely used for existing-structure data acquisition for 3D indoor scenes. However, semantically rich 3D indoor model creation of building interiors using scanned point clouds encounters critical difficulties: the complex design of indoor structures, not to mention obstacles, requires time-consuming manual operation and thus large data sizes, which often leads to system slowdown or even complete failure.

In current practice, the creation of accurate 3D indoor models is largely an interactive process performed by service providers who are contracted to scan and model a facility [20]. A project might require several months to complete, depending on the complexity of the facility and modeling requirements [4]. The basic geometric modeling process for manual creation of a 3D indoor model involves fitting geometric primitives to the 3D point cloud directly. Existing commercial software typically includes tools for fitting geometric primitives such as planes, cylinders, spheres, and cones to the point cloud [21,22]. These tools are semi-automated and require significant user input. For example, to model a planar surface, the user selects a few points or a patch of data, and a plane is fitted to the selected data [23]. In this way, approximate boundaries of the patch can be identified, but in practice, these boundaries can be irregular and inaccurate. To improve the efficiency and accuracy of the modeling process, some researchers have proposed semi-automatic modeling methods through using cross-sections and surface extrusion algorithms. A semi-automatic methodology for improved productivity in as-built BIM creation with respect to large and complex indoor environments is proposed in [24]. In this approach, the plane feature and remaining points are automatically extracted from the point cloud and used as a reference to facilitate the manual construction of the as-build BIM. [15] present similar work. Their method generates the BIM model by importing a 3D wireframe model into BIM tools for manual as-build modeling.

Because the same types of primitives must be modeled throughout a facility, the steps are highly repetitive and tedious [25,26]. In addition, due to the complexity and uniqueness of specific buildings, modelers must be highly skilled and proficient. Even with training, decisions about exactly what elements to model and how to model them are sufficiently subjective to cause significant variability in the quality of the models produced by different personnel. To manage this problem, some recent studies have looked to full automation. Okorn et al. presented an automated method to create accurate 2D floor plane models of building interiors from a 3D point cloud [27]. This method involves creating a histogram of height data to detect floor and ceiling data, and then extracting the line segments corresponding to walls based on the point density of the projected remaining points. Budroni and Boehm developed a volume sweep and cell decomposition method for the reconstruction of Manhattan-world interior scenes from point clouds [28]. The output is a 3D CAD model of indoor environments, but the key assumption is the absence of furniture or small objects. Moreover, this method is only applied in a single room or space and the openings are not taken into consideration. In the approach proposed by Sanchez and Zakhor, points are classified into floor, ceiling, wall, and remaining points using point normal orientations [29]. The planar patches of floors, ceilings, and walls are fitted and the boundaries are estimated using alpha shapes. Adan and Huber propose a 3D reconstruction method for interior wall surfaces under high levels of occlusion and missing data [30]. They recover occlude parts of reconstruction surfaces and perform opening detection using a Support Vector Machine (SVM) algorithm. Using knowledge of different types of elements and the contextual relationships between them, Xiong et al. extend this approach so that it is capable of identifying and modeling the main visible structural opening detection components of an indoor environment including walls, floors, ceilings, windows, and doorways [2]. Ochmann et al. proposed a volumetric, parametric building model by also incorporating contextual information [14]. A global optimization approach was used to reconstruct of wall elements shared between rooms while simultaneously maintaining plausible connectivity between all wall elements. Mura et al. first proposed an occlusion-aware method to extract candidate wall elements while taking into account possibly occluded parts of the surfaces to determine real wall heights for filtering out invalid candidates [9]. The projected 2D lines of candidate wall patches corresponding to the same wall structure were merged and a 2D cell complex was obtained from the intersection of the remaining lines. By computing the diffusion distances between faces, a global measure of affinity was used to iteratively cluster the cell complex into individual rooms. However, this method does not attempt to recognize opening structures and the relationships between these indoor elements. 

The above discussion shows that the constraints of automatic indoor reconstruction are due to either missing data due to occlusions and clutter, or a lack of robust space partitioning. Because consumer-level 3D acquisition devices such as RGB-D sensors allow more convenient interoperates and are capable of obtaining additional valid information such as camera trajectories, frame sequences with timestamps, and colorized point clouds that provide new opportunities to integrate multiple types of data for indoor reconstruction. Li et al. proposed a robust CPU-based approach to reconstruct indoor scenes with a consumer RGB-D camera. The method combines feature-based camera tracking and volumetric-based data together. The experiment results demonstrate a good reconstruction performance in terms of both robustness and efficiency [31]. Similarly, a robust approach to elaborately the indoor environment with a consumer depth camera is proposed and the main contribution of this research is using the local-to-global registration to obtain complete scene reconstruction [32]. Besides, to enhance the robustness of RGB-D mapping, a multi-camera dense RGB-D SLAM system is proposed and the experimental results shows the multi-camera system is able to increase the efficiency and improve the accuracy significantly [33]. Chen at.al uses a method to automatically model indoor scenes based on low-quality RGB-D sequences by establishing the relationships between objects and knowledge from the model database [34]. Based on large-scale RGB-D datasets, Dai et al. use the network architecture of the 3D Network-in-Network [35] for 3D object classification and conducts semantic labeling by extending the semantic segmentation method [36] to 3D [17]. However, these two methods focus on the reconstruction of indoor furniture rather than indoor structural components. Systematic indoor reconstruction methods based on RGB-D mapping systems are rarely investigated.

In this study, we seek to establish a semi-automatic indoor reconstruction method by incorporating multiple types of data from low-quality RGB-D sequences. This paper presents the following novel findings. First, the proposed indoor reconstruction makes full use of the camera trajectory and semantic labeling tags of RGB-D frames except colorized point cloud, thus improving robustness and recognition accuracy for indoor reconstruction. Second, inevitable mapping errors can result in inconsistency between the planes detected from different functional spaces, for instance the joint wall between two spaces, an opening and its connected wall. A global optimization approach is designed to eliminate inconsistency between the adjacent walls, walls, and openings and maintain plausible connectivity between the walls and the relationships between the walls and openings.

## 3. Methodology

### 3.1. Overview

The RGB-D mapping system used in this research contains two sensors: one RGB camera, and one IR sensor. The IR sensor is combined with an IR camera and an IR projector. This kind of sensor system can be highly mobile, and attached to an iPad, iPhone, or other mobile instrument. Unlike the traditional laser scanning system, the system captures 640 × 480 registered RGB images and depth images at 30 frames per second, which is convenient for interactive RGB-D frame labeling. Figure 1 shows its hardware structure and the observed depth and RGB frames with timestamp.

We next describe our methods to automate the identification and reconstruction of accurate and consistent indoor components from low-quality RGB-D sequences as shown in Figure 2. The main idea behind our approach is to extract indoor components by making full use of the information from RGB-D sequences, which contains the camera trajectory, labeling information, and low-quality point clouds. Because RGB-D frames are labeled by camera position and orientation, timestamp, and start-stop point associated with RGB-D sequences, the camera trajectory and the RGB-D sequences are divided into several subsections according to the range of each space and the positions of openings. Next, the point cloud of each subsection can be obtained by merging the corresponding RGB-D frames. Note that all point clouds to be processed are generated after applying the depth calibration model, and data preprocessing operations are applied to each point cloud. 

Two different processing flows are used for individual space reconstruction and opening reconstruction, respectively: (1) For space extraction, a region growing plane segmentation method is first used for plane detection. Based on the normal of the recognized planes, the surfaces are organized in vertical and horizontal directions. Each plane is described using a cluster of points and plane parameters. We distinguish the types of planes by following the classification rules, and wall candidates are derived from vertical surfaces observed in the scans. Subsequently, an interior element filtering algorithm is used to separate the wall planes from the vertical planes. Finally, space layout is generated based on a boundary generation algorithm, which considers the intersection of all wall candidate centerlines in the horizontal plane. (2) Rather than detecting the opening by finding the holes of the triangulated model of the wall planes, we propose a new noise robustness method based on the properties of convex hull, octrees structure, Euclidean clusters and the camera trajectory for opening generation, which is inapplicable to the data collected in the indoor environments due to the inevitable occlusion. A global optimization approach is designed to eliminate inconsistency between adjacent walls and maintain plausible connectivity between the walls and the relationships between the walls and openings. Finally, the final model is stored according to the CityGML3.0 standard.

### 3.2. Data Pre-Processing

The point cloud data obtained by the RGB-D mapping system usually contain noise and varying point densities in different regions caused by measurement errors and high-frequency data streams. This complicates the estimation of local point cloud characteristics such as surface normal or curvature changes. Figure 3a shows a raw point cloud generated from RGB-D system. A sparse outlier removal algorithm [37] is used to distinguish and remove the isolated points from the original point cloud. The sparse outlier removal module corrects these irregularities by computing the mean μ and standard deviation σ of nearest neighbor distances. The neighbor points are defined by k, which represents the number of points to use for mean distance estimation. On the premise that the distances are random values with a Gaussian distribution, we trim the points that fall outside the μ ± α∙σ. The value of α depends on the size of the analyzed neighborhood. In our implementation, we set α = 1 and k = 50, because experiments with multiple RGB-D datasets have confirmed the applicability of the μ ± α thresholds, with approximatively 1% of the points considered to be noise (Figure 3b). Data redundancy and density inconsistency problems can occur when all RGB-D sequences are merged directly.

As a result, a voxelized grid algorithm was used to down-sample the point cloud, which is able to unify the point densities of the whole scenes and speed up data processing. The voxel grid structure creates a 3D voxel grid over the input point cloud data with a specific parameter β. The value of β depends on the size of the voxel grid. Each voxel has its own specific boundary according to the setup size. After they are placed in their corresponding voxels, all the points present in the same voxel are removed and a centroid point for the point group is created (Figure 3b,c). Thus, the larger the voxel, the more points are eliminated.

### 3.3. Spaces Division and Extraction

#### 3.3.1. Spaces Partition

An RGB-D mapping system uses short range distance measurements and high-frequency data collection, which facilitates the semantic labeling for the RGB-D frames during scene scanning. RGB-D datasets consist of a series of RGB-D frames and each frame is localized by the camera pose and timestamp. Therefore, rather than distinguishing room spaces based on the detected wall planes, we interactively define the start-stop position for each functional space (e.g., room, corridor, kitchen, etc.). Thus, the RGB-D sequences are divided into several subsections associated with the camera trajectories. And the corresponding point cloud for each functional spaces can be obtained through merging all frames in each start-stop section. Similarly, the frames on the start-stop positions and their adjacent frames of each functional space are generally recognized as the frames containing door components. The point cloud containing doors can be extracted through merging the point clouds of all adjacent frames. Figure 4 shows sample data of the labeled RGB-D sequences. Each functional space is described by a series of RGB-D frames, a subsection of camera trajectory. Different colors in the camera trajectory belong to different functional spaces. Based on the camera pose and the RGB-D frames, a colorized point cloud of each space can be recovered.

#### 3.3.2. Generation of Wall Candidates

The recognition of wall planes is a prerequisite for space reconstruction. A region growing plane segmentation method is used for plane detection. This algorithm merges points that are close enough in terms of the smoothness constraint. The work of the algorithm is based on the comparison of the angles between points normal. The points {P} are first sorted by their curvature value and the algorithm picks up the point with minimum curvature value and starts the growth of the region, because the point with the minimum curvature is always located in the flat area. In our solution (1) to estimate the normal of each point, the distribution of neighbor points is used and the principal direction is estimated through plane fitting with the least square algorithm; (2) to estimate the curvatures at each point on a discrete 3D point cloud, the distribution of neighbor points are also used. The main steps of this method included using the estimation of normal section lines for normal curvature and the optimization of all these normal curvatures. The principal curvatures and principal directions are estimated through the least square fitting of all normal curvatures related to all neighbor points. The picked point is added to the set called seeds $\left\{S_{c}\right\}$. For each seed point, the algorithm finds neighboring points $\left\{B_{c}\right\}$ and calculates their normals. If the angle between the normal of the seed point and the normal of the neighboring point is less than the defined angle threshold $\theta_{th}$, the current point is added to the current region. After that, each neighbor is tested for the curvature value. If the curvature is less than threshold value $c_{th}$ then this point is added to the seeds. Meanwhile, current seed is removed from the seeds. The seeds container iteratively grows until it is empty. If the seeds set becomes empty this means that the algorithms has grown the region and there is no new candidate seeds. The process is repeated from the beginning. As shown in Figure 5a, the output of the segmentation method is a set of segmented point cloud clusters.

Based on the normal of the recognized planes, the surfaces are organized in the vertical and horizontal directions. Each plane is described by a cluster of points and plane parameters. We distinguish the types of planes by following the classification rules. Wall candidates are derived from vertical surfaces observed in the scans. In the first step, two constraints are used to determine the potential wall surfaces (Figure 5b): (a) The angle between the normal of the plane and the vertical direction is less than the defined normal threshold $n_{th}$. (b) The maximum of the plane length is larger than the sufficiently large length $l_{m}$. The plane length means the dimension of a plane, which can be calculated from the envelope of the point cloud. Similarly, the potential floor plane and ceiling plane can be identified based on the normal constraint and the height constraint (Figure 5c). In the second step, all of the horizontal planes are projected to the floor plane and a convexmhull algorithm is used to extract the 2D boundary of the horizontal planes. Figure 5d shows a sample of boundary extraction of horizontal planes. In the third step, to filter out the vertical planes of the interior elements, all of the vertical planes are projected to the floor plane. Therefore, we encounter the problem of determining the inclusion of dozens of point cloud clusters {P} in a 2D planar polygon. 

An interior elements filtering algorithm is used to separate the wall planes from the vertical planes (Algorithm 1). For each point cloud cluster Pi, we adopt the RANSAC line fitting method to obtain the optimal fitted line L|{y=ax+b}. The RANSAC line fitting method iteratively computes the optimal line by minimizing the deviations R2 of a set of points, which are picked randomly (Equation (1)). The fitted line segment consists of two endpoints {psti,pendi}. To determine the inclusion of a fitted line segment in the 2D boundary polygon, the cross number method is applied. This method counts the number of times a ray starting from a point crosses a polygon boundary edge separating its inside and outside. If this crossing number cn is even, then the point is outside. If the crossing number cn is odd, the point is inside. In this paper, three kinds of situations are considered. (a) When the crossing number of two endpoints are odd, the fitted line segment is inside a 2D polygon boundary. It means that the corresponding point cloud cluster is from the interior elements (Figure 6a); (b) when the crossing number of two endpoints are even, the fitted line is outside the 2D polygon boundary. Thus, the corresponding point cloud cluster is taken from the wall components (Figure 6b); (c) by considering the measurement errors in a low-quality RGB-D point cloud, we can encounter a situation where one endpoint is inside and one endpoint is outside. An intersection point piters between the fitted line and the polygon boundary can be obtained. The distance {dst,dend} from each endpoint {psti,pendi} to the intersection point piters can be calculated. When the percentage of the distance obtained from the outside endpoint and the distance obtained from the inside endpoint is higher than the defined threshold perl, we determine it to be a wall plane. Otherwise, we determine it to be an interior element (Figure 6c):
(1)R2=∑i=0n[yi−f(xi,a,b)]2Where f(xi,a,b)=axi−b
**Algorithm 1: Interior elements filtering algorithm****Input:**Projected potential wall planes:  {P}={Pi|1≤i≤N}2D polygon consist of points: Poly= {pnj|1≤j≤M}Percentage threshold of length in interior and external of polygon: perlLine fitting function based RANSAC method: F(.)Fitted line: L={psti,pendi}Distance calculation function: D(.)Distance from endpoints to intersection point: dst,dendIntersection function of two line segments: L(.)Intersection point of two line segments: pitersCrossing number function to determine the inclusion of a point in a 2D polygon: Ω(.)Crossing number from a ray of endpoint to the polygon boundary edges: cnst,cnend**Output:** index of interior planes: {indexIA}={id|1≤i≤N}; index of wall planes: {indexEA}={id|1≤i≤N};1. indexA=∅,L=∅2. **While**
{P} is not empty **do**
3.  $cn_{st}=0$,$\;cn_{end}=0$
4.   Do line fitting for $P_{i}$
5.   $L\leftarrow F\left(P_{i}\right)$
6.   Detection the crossing number of the ray from the endpoint and the polygon boundary 7.  $cn_{st}\leftarrow \Omega \left(p_{st}^{i},\mathrm{Poly}\right)$,$\;cn_{end}\leftarrow \Omega \left(p_{end}^{i},\mathrm{Poly}\right)$
8.  **If**
$cn_{st}$ is odd && $cn_{end}$ is odd **then**
9.    {indexIA}←{indexIA}∪i
10.  **else if**
$cn_{st}$ is even && $cn_{end}$ is even **then**
11.    {indexEA}←{indexEA}∪i
12.   **else**
13.   Do intersection for the fitted line and the polygon boundary 14.   $p_{iters}\leftarrow L\left(l\right)$
15.    Calculate the distance between endpoints ($p_{st}^{i},p_{end}^{i}$) and $p_{iters}$
16.   $d_{st}\leftarrow D\left(p_{st}^{i},p_{iters}\right)$,$\;d_{end}\leftarrow D\left(p_{end}^{i},p_{iters}\right)$
17.   **if** ($cn_{st}$ is even && $\frac{d_{st}}{d_{end}}>per_{l}$) || ($cn_{end}$ is even && $\frac{d_{end}}{d_{st}}>per_{l}$) **then**
18.    {indexEA}←{indexEA}∪i
19.   **else**
20.    {indexIA}←{indexIA}∪i
21.   **end if**
22.  **end if**23. **end while**24. **Return**
{indexIA},{indexEA}

#### 3.3.3. Determination of Space Layout and Parameterization

As described in Section 3.3.2, the point cloud cluster of wall planes is generated. Due to measurement distance limitations, clutter and occlusion, the detected wall planes can create issues of incompleteness, therefore, we propose a boundary generation algorithm for space layout determination by considering the intersection of all wall candidate centerlines in the horizontal plane. Note that this algorithm is designed for situations where the spaces consist of straight walls (Algorithm 2). 

First, we use the RANSAC line fitting method to obtain the optimal fitted line {L} for all 2D projected walls, as detailed in Section 3.3.2. For each line, two nearest neighbor lines $\mathrm{NN}p_{st},\mathrm{NN}p_{end}\;$and the two corresponding endpoints are obtained by calculating the minimum distance between the current endpoint and the endpoints of others. To organize the line segment by order, the algorithm starts from the first line and iteratively adds the index of the nearest neighbor line segment into the vector of lines segment index {Ls}. Meanwhile, the corresponding intersection points are calculated and added into the vertex container $\left\{p_{iters}\right\}$. It continues until the algorithm returns back to the starting line. Most of the vertices of the boundary can be generated by line intersection operations. However, some parts of walls might be missed during scanning, which results in disjoint relationships the between two adjacent lines. Figure 7. shows two different situations during boundary vertex determination. In Situation 1 (Figure 7b), two lines are almost orthogonal, and the intersection point can be easily obtained using a line intersection operation. In Situation 2 (Figure 7c), because the connecting wall between line1 and line2 is missing, the two line candidates are almost paralle. The algorithm addresses this situation by checking the normal angle of two lines $C\left(L^{{\mathrm{Ls}}^{i}},L^{\mathrm{tarL})}\right)$. If the normal angle $C\left(L^{{\mathrm{Ls}}^{i}},L^{\mathrm{tarL})}\right)$ is less than angle threshold $\theta_{th}$, then we added the endpoint of the line segment into the vertex container. In our experiments, we set $\theta_{th}=30^{\circ }$. 

Finally, based on the boundary generation algorithm, the space layout boundary can be obtained, which consists of the intersection points and the endpoints of segmentation. The space can be parameterized according to the height calculated from floor and ceiling planes.


**Algorithm 2: Boundary generation algorithm**
**Input:**Projected wall planes: {P}={Pi|1≤i≤N}Fitted lines vector: {L}={psti,pendi}Line fitting function based RANSAC method: F(.)Minimum Distance calculation between the current point and others function: D(.)Enum of the endpoints of line: eu=(st, end)Map of Index of Nearest Neighbors of the fitted lines and the endpoints of the target line: Mpst{NNpst, euEP  },Mpend{NNpend,euEP}Index of the lines by order: {Ls}Intersection function of two line segments: L(.)Angle calculation of two line function: C(.)Angle threshold: θth**Output:** Vertex of space boundary: {piters}1. L=∅, Dmin=1000, Lcur=0,Ltar=12. **for** i = 0 to size ({P}) **do**
3.  Do line fitting for Pi
4.  {L}←F(Pi)5. **end for**
6. **for** i = 0 to size ({L}) **do**
7.  Calculate minimum distance between the endpoint psti or pendi and others and output the index of nearest neighbor. 8.  {NNpst}←D({L}), {NNpend}←D({L})9. **end for**10. Set the first line as start line: 0→{Ls}11. **for** i = 0 to size ({L}) **do**
12.  **if**
$Mp_{eu}\left(\mathrm{NN}p_{eu}{}^{{\mathrm{Ls}}^{i}}\right)$ equal to 0 13.    tarL=0
14.  **else**
15.    $\mathrm{tarL}=Mp_{eu}(\mathrm{NN}p_{eu}{}^{{\mathrm{Ls}}^{i}}$) 16.  **if**
{Ls} don’t contains tarL
17.  **Then**
18.    tarL→{Ls}
19.    Calculate the intersection point of two adjacent line and add to points of space boundary 20.     $\mathrm{tempP}\leftarrow L\left(L^{{\mathrm{Ls}}^{i}},L^{\mathrm{tarL}}\right)$
21.    **If** the angle $C\left(L^{{\mathrm{Ls}}^{i}},L^{\mathrm{tarL})}\right)$ is less than $\theta_{th}$
22.     $p_{eu}^{\mathrm{tarL}}\to \left\{p_{iters}\right\}$, $p_{Mp_{eu}\left(\mathrm{euEP}\right)}^{{\mathrm{Ls}}^{i}}\to \left\{p_{iters}\right\}$, 23.    **else**
24.     $\mathrm{tempP}\to \left\{p_{iters}\right\}$
25.    **break**
26.  **end if**
27.  **if**
tarL is equal to 0 28.    **break**
29.  **end if**
30. **end for**31. **Return**
$\left\{p_{iters}\right\}$

### 3.4. Opening Extraction

Because the mobile mapping system needs to enter and exit the rooms, we assume that the doors are opened during the data collection process. Thus, the opening detection problem can shift to how to find the vacancy in the point clouds. Traditionally, most researchers detect the opening by finding the holes of the triangulated model of the wall planes. This approach is unsuitable for data collected in indoor environments due to inevitable occlusion. In this paper, we propose a new noise robustness method based on the properties of convex hull, octrees structure, Euclidean clusters, and the camera trajectory. Figure 8 shows the workflow for generation of the opening components.

Because we interactively define the start-end positions for each functional space when entering or exiting doors, we obtain the specific RGB-D frames containing door components from the sequences of functional spaces, as shown in Figure 8. Based on the camera pose and the frame sequences, a colorized point cloud for each door component can be recovered. A plane segmentation method is first used for plane detection and the planes containing doors are derived from vertical planes. 

To detect the opening from the segmentation planes, a three-step strategy is involved. We first compute the convex hull polygons for the given point cloud ${\mathrm{Ps}}_{d}$, and create a new set of point cloud ${\mathrm{Ps}}_{\mathrm{ch}}$ by filling the polygon region with evenly distributed points. Then, the octree structures are constructed for ${\mathrm{Ps}}_{d}$ and ${\mathrm{Ps}}_{\mathrm{ch}}$ respectively. With the assumption of vacancy in opening, K nearest neighbor searching is used for detecting the vacancy region between two octree structures and the corresponding point cloud ${\mathrm{Ps}}_{\mathrm{inc}}$ are obtained. Finally, Euclidean cluster extraction algorithm is used to divide ${\mathrm{Ps}}_{\mathrm{inc}}$ into separated components. The details are descripted as follows:

First, the point cloud containing openings is projected to the best fitting planes of itself. A convex hull algorithm is then used to compute the envelope of the projected planes containing openings. As shown in Figure 8a,b, the algorithm generates convex polygons to represent the area occupied by the given points. The envelope of the detected convex hull is subsequently filled by the evenly distributed points and the point cloud ${\mathrm{Ps}}_{\mathrm{en}}$ is generated (Figure 8c). Since the envelope is larger than the original convex hull, the cross number method detailed in Section 3.3.2 is used to determine the inclusion of the point in the convex hull polygon and only the point inside the polygon is added into the new point cloud ${\mathrm{Ps}}_{\mathrm{ch}}$. Figure 8b shows the convex hull polygon as a red solid line, the envelope of the convex hull as a blue dashed line, the points inside the polygon as green dots, and the points outside the polygon as blue dots. 

Second, as shown in Figure 9, an octree structure with the same leaf size is generated for two point clouds. Therefore, the problem of opening detection shifts to finding the differences between the octree structure of the current opening planes ${\mathrm{Ps}}_{d}$ and the filled point cloud ${\mathrm{Ps}}_{\mathrm{ch}}$. Note that an appropriate leaf size is determined according to the density of the current opening planes. The vacancy in the current door plane can be found via K nearest neighbor searching on two octree structures. Figure 9c shows the schematic diagram of the point cloud ${\mathrm{Ps}}_{\mathrm{inc}}$ after changing detection. Subsequently, the Euclidean cluster extraction algorithm is used to divide the point cloud ${\mathrm{Ps}}_{\mathrm{inc}}$ into separated opening components (Figure 8e). Finally, door components are reconstructed by the maximum of the point cloud ${\mathrm{Ps}}_{\mathrm{inc}}$ and the 2D fitted line of each opening component (Figure 8f). Based on the proposed method, window and door components can be reconstructed based on the point cloud of tagged RGB-D frames.

### 3.5. Global Optimization for Planes

Even when a high-precision calibration method is applied to improve the mapping accuracy of the RGB-D system, it inevitably causes system and mapping errors in the point cloud generated from RGB-D sequences. This can result in inconsistencies between the planes detected from different functional spaces, such as a joint wall between two spaces, an opening, and its connected wall. Therefore, a global optimization approach is used to eliminate the inconsistency between the adjacent walls, and walls and openings, and maintain the plausible connectivity between the walls and the relationships between the walls and openings. 

Normally, two adjacent spaces often share the same plane (called the global plane in this paper), which can be extracted from the whole model instead of individual functional spaces. The whole model is generated by merging all RGB-D sequences, and the global planes can be obtained by applying a plane segmentation algorithm to the whole model. Because wall planes are detected from the individual point cloud of each space, there can be significant discrepancies between the global planes and the corresponding wall planes. As shown in Figure 10, wall plane1 $wP_{1}$ and wall plane2 $wP_{2}$ share a same global plane $gP_{1}$, so it is hard to guarantee that they will have the same plane parameters during plane segmentation. To eliminate inconsistencies between the wall planes and global planes, all of the wall planes sharing the same global plane are projected onto the corresponding global plane. 

To find the corresponding global plane for each wall plane, we first calculate the angle of the plane normal $\theta_{diff}$ between the specific wall plane according to Equation (2) and the planes detected from the whole model. Once $\theta_{diff}$ are less than the threshold$\;\theta_{th}$, the global plane candidates are filtered out. We define the wall plane wP, $\left\{a_{w}x+b_{w}y+c_{w}z+d_{w}=0\right\}$, and the global plane candidates $\mathrm{gP},\;\left\{a_{g}x+b_{g}y+c_{g}z+d_{g}=0\right\}$. To find the optimal global plane, distance $d_{diff}$ between the wall plane and each global plane candidates are calculated. Because the distance between two intersecting planes is 0, two compared planes are forced to have the same plane normal, which means that the plane equation of wall plane wP becomes $\left\{a_{g}x+b_{g}y+c_{g}z+d_{w}=0\right\}$. Therefore, distance $d_{diff}$ can be calculated according to Equation (3). The optimal global plane is found when the minimum value of the distance $d_{diff}$ is obtained. As shown in Figure 10, $wP_{1}$ and $wP_{2}$ share the same plane $gP_{1}$, and $gP_{2}$ is the global plane of $wP_{3}$. Similarly, the connected wall planes of each opening can be detected and the plane parameters of corresponding opening are corrected, as shown in Figure 10:(2)$$\theta_{diff}=\mathrm{acos}\left(n_{wall}\cdot n_{door}/||n_{wall}||||n_{door}||\right)<\theta_{th}$$
(3)$$d_{diff}=\left|d_{g}-d_{w}\right|/\sqrt{a_{g}{}^{2}+b_{g}{}^{2}+c_{g}{}^{2}}$$

## 4. Experimental Results and Discussion

We tested our proposed methodology on synthetic multi-level data sets, four measurement data sets collected in a single room and one data set acquired from a space with complex layout. All of the datasets were collected using the RGB-D mapping system shown in Figure 11a and samples of the RGB image and depth image are plotted in Figure 11b,c. In our RGB-D system, Kinect sensors are mounted on NVIDIA Jetson TX2 and carried by a trolley. A RGB-D SLAM method presented by our previous work [8] is used for camera tracking and pose optimization, which enable to obtained accurate camera pose for each frame and the corresponding point cloud. To facilitate the identification of the start-end position of functional space interactively, the RGB-D sequences are endowed with specific tags through responding to the key board message. Therefore, each data set contains the colorized point cloud and the RGB-D sequences associated with timestamp, camera position and labeling tags. For each data set, sparse outlier removal and down-sampling algorithm are used to reduce the density and remove the noise of the raw colorized point cloud. To quantify the reconstruction results, two kinds of error metrics are used. The first is the accuracy of the quantity of the extracted components. The second is the accuracy of the area dimension. 

In the first case study, the data set contains 115 RGB-D frames. As shown in Figure 12a, because the windows are sheltered by a curtain, only the frames containing door components are labeled. From the view of the point cloud, most parts of the room are scanned and modeled. The room contains six walls and one door components. As shown in Figure 12b, the raw colorized point cloud was first segmented into a set of plane clusters. The planes are classified into vertical planes and horizontal planes based on the normal of the planes, and the wall candidates were distinguished by following the classification rules of the plane normal. In this experiment, we set the normal threshold $n_{th}=5{}^{\circ}$ and a length threshold $l_{m}=1\;m$. Subsequently, the interior elements filtering algorithm is used to separate the wall planes from the vertical planes and six wall planes can be extracted, as shown in Figure 12b. To obtain the wall planes connected to the door planes, we set the distance threshold between two planes $d_{\mathrm{diff}}=5\;\mathrm{cm}$ and the angle threshold of plane normal $\theta_{diff}=2{}^{\circ}$. In Figure 12b, the door plane is projected to the connected wall plane and the point cloud of door component coloring in green is correctly extracted using the opening extraction method outlined in Section 3.4. Figure 12c shows the skeleton of the reconstructed components. Six walls, one door, one floor component, and one ceiling component are recognized from the data set. The relationship between the components is also correctly recovered. Based on the parameterization results, the components are saved according to the CityGML3.0 standard, as shown in Figure 12d. Recognized accuracy is measured to evaluate the performance of the component reconstruction (Table 1). In this case study, all recognized components are correctly categorized and reconstructed. The area dimension of the recognized components are also compared with the manually measured area dimensions from the point cloud. The absolute difference is calculated for each categorized component. 

Table 2 shows the comparison of results between the recognized dimension and measured dimension of each type of component. The door category achieves the most accurate results because of the use of specific frames. As expected, the walls, ceilings, and floors generate similar results and achieve lower accuracy due to the deficiencies of the raw point cloud data. 

To further validate the robustness of the proposed methodology, two more case studies are conducted. Figure 13. shows the reconstruction process for case study (2), which is a single room with more complex layout comparing with case study (1). Two frames are labeled with the tag “door”. Due to the limitation of the view angle, only the bottom of the room is scanned. Similar to case study (1), the reconstruction results for wall and door components are illustrated in Figure 13c, and the corresponding CityGML modeling is shown in Figure 13d. In case study (2), 13 walls and two doors are recognized directly from the raw data set. Due to the occlusion problem during data collection, a wall component is missing when wall candidates are generated, as shown in Figure 13c (bottom). As expected, the missing wall is recalled based on the rules of wall determination noted in Section 3.3.3. The algorithm constructs a new line when the normal angle of two adjacent lines is less than the angle threshold $\theta_{th}$ = 50 and a new wall is reconstructed in Figure 13c (top). The evaluation of the extracted components of case study (2) is shown in Table 1. It achieves 100% accuracy in component reconstruction in this situation. As shown in Table 2, case study (2) generated similar results. The absolute errors of recognized dimension were all within 2%, and the door component achieved the best result.

In case study (3), the tested building has a more complicated structure containing 1278 RGB-D frames, six functional spaces, dozens of door components, and several window components Figure 14a (1) shows the raw data set with camera trajectory and sample frames containing doors and windows. Six functional spaces are segmented and reconstructed successfully according to the tags of RGB-D frame, shown in Figure 14a (2). Figure 14a (3) and (4) show the skeleton of the whole model and the CityGML model of the scenes. Based on the evaluation results shown in Table 1, two of 28 walls, two of 25 doors and one of nine windows are not successfully recognized from the point cloud data, and the proposed reconstruction method achieves recognized accuracy of 89%, 92%, and 88% respectively. One wall is recalled based on the rules of wall determination. Figure 14b lists the reconstruction results of each functional space. In the reconstruction results of Spaces 3 and 4 in Figure 14b, two recognized door components marked with red borders contain more than one door entity, which results in a lower number of recognized door components. Similarly, Figure 15 shows the reconstruction results of case study (4), which contains 857 RGB-D frames, five functional space and several openings. Raw data associated with camera trajectory and the reconstruction results are presented in (a). Figure 15b list the reconstruction results of each functional space. As shown in Table 1, only one of 23 doors is not successfully recognized and it achieves 95 % recognizing accuracy. 

Table 1 lists time consumption and the measurement accuracy of reconstructed components. For the time consuming, the proposed method costs 23.2 s, 31.8 s, 84.8 s, 78.3 s for components reconstruction in case (1), case (2), case (3) and case (4), respectively. The processing time increases with the complexity of the scenes. In case (3) and case (4), the algorithm achieve the accuracy ranging from 97% to 100% in all recognized component categories. The door and window components achieve the best results, and this finding is consistent with the conclusions of case study (1) and case study (2). To validate the effectiveness of the proposed, the reconstruction results are compared with the state-of-art method proposed by Wang et al. [1], which was used for BIM extraction with laser point cloud and mainly concentrated on the building with single functional space. As demonstrated in their experimental results, the proposed method by Wang et al. is able to achieve an average measurement error with 89.094, 95.25% and 92.376 in three different kinds of building respectively. In horizontal comparison, the proposed method is used for reconstruction in single functional space, and achieves 97.23%, 98.68% measurement accuracy respectively shown in Table 2. It indicates that the proposed method provided better reconstruction accuracy than Wang’s method. Besides, for case (3) and case (4), the proposed method achieve about 98.21% and 97.06% measurement accuracy, which are also better than the reconstruction accuracy presented by Wang et al. [1]. 

In addition, we plot the average dimension error of each components in four study cases in Figure 16. The average error for each components is calculated by dividing whole error by measured area dimension. As shown in Figure 16, opening components achieve higher accuracy than wall, ceiling and floor in all study cases. The possible cause of this is that point clouds are usually difficult to comprehensively collect from large spaces due to the limitations of mapping range or occlusion.

## 5. Conclusions

In this paper, we propose an automatic indoor reconstruction methodology using low-quality RGB-D sequences. Our approach allows for the robust generation of semantically rich 3D indoor models and demonstrates applicability and reconstruction power for complex real-world datasets. From our theoretical analysis and experimental validation, the following conclusions can be drawn: 

Benefiting from the multiple types of data set and the advantage of interactive data collection of the RGB-D mapping system, the proposed method provides new opportunities to use low-quality RGB-D sequences to reconstruct semantically rich 3D indoor models that include wall, opening, ceiling, and floor components. For point cloud data with significant occlusion, most components can be recognized correctly to achieve an average accuracy of 97.73%. Some components in case study (2) and case study (3) that are absent from the point cloud can be recalled based on the layout determination algorithm. The reconstruction results indicate the robustness of the proposed methodology for low-quality point clouds.The proposed reconstruction method produces an area dimension error within 3% for all cases. The measurement results indicate that modeling accuracy can be affected by the range sizes of the components. Higher range sizes result in lower accuracy.

The automatic reconstruction method based on low-quality RGB-D sequences discussed here enables one to take full advantage of the information and the mode of data scanning provided by the RGB-D mapping system. This provides a fast, more convenient, and lower-cost solution for semantically rich 3D indoor mapping. The next step in this research will to be improve the methodology by introducing algorithms to deal with complex shapes such as cylinders, curved surfaces and so on, which would make the method more robust when modeling more complicated indoor scenes. 

## Figures and Tables

**Figure 1 sensors-19-00533-f001:**
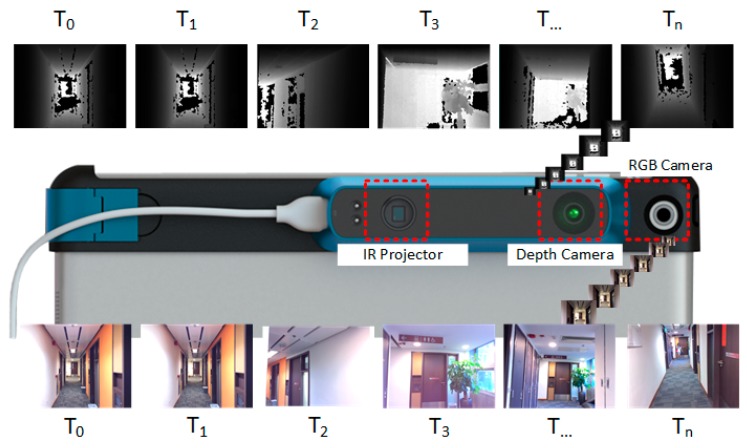
Hardware scheme of the RGB-D sensor and the acquired RGB-D sequences.

**Figure 2 sensors-19-00533-f002:**
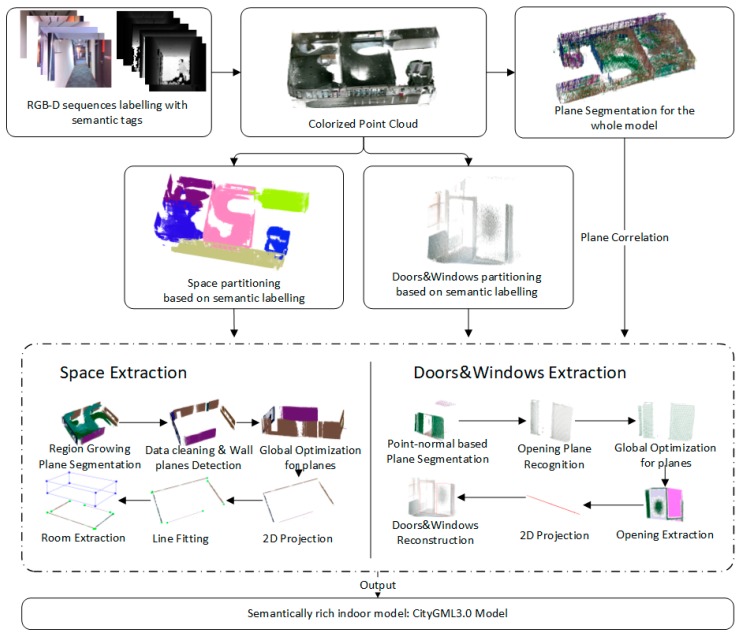
Framework of indoor reconstruction from low-quality RGB-D sequences.

**Figure 3 sensors-19-00533-f003:**
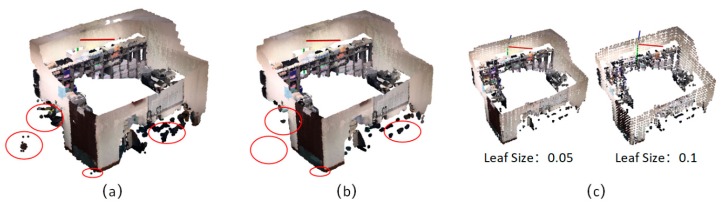
A sample of data pre-processing, (**a**) raw point cloud, (**b**) point cloud after outlier removal operation, (**c**) point cloud after down-sampling with different voxel size.

**Figure 4 sensors-19-00533-f004:**
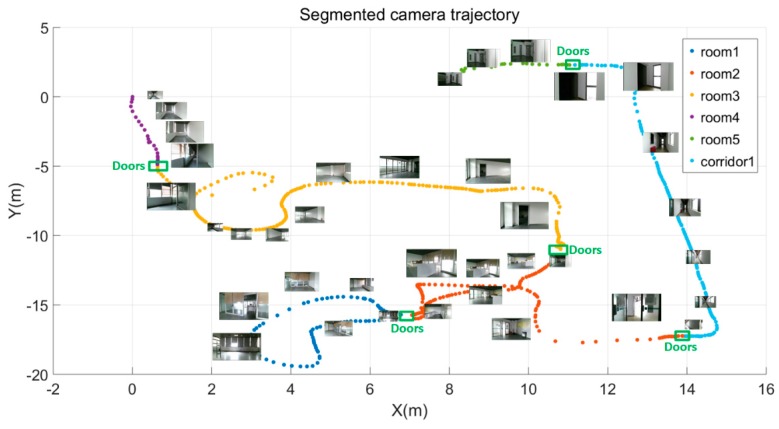
Data of RGB-D sequences.

**Figure 5 sensors-19-00533-f005:**
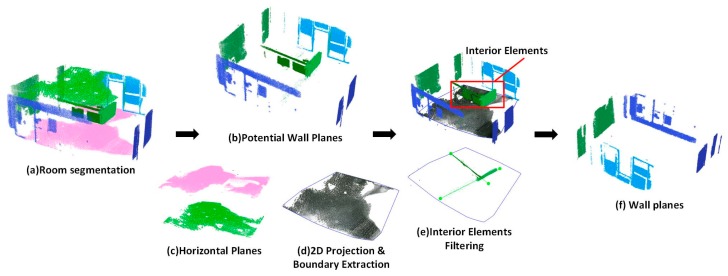
Sample of generation of wall candidates.

**Figure 6 sensors-19-00533-f006:**
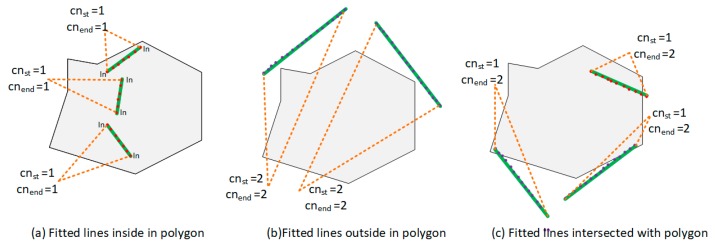
Crossing number method for wall planes extraction.

**Figure 7 sensors-19-00533-f007:**
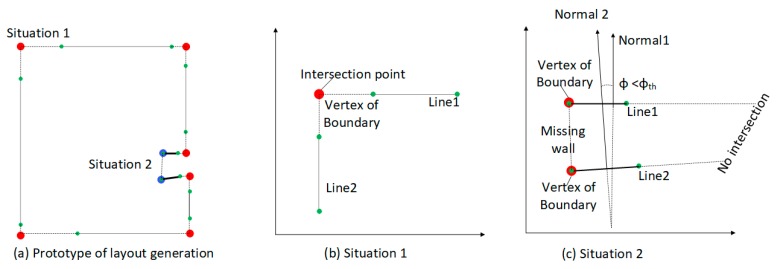
Determination of space layout.

**Figure 8 sensors-19-00533-f008:**
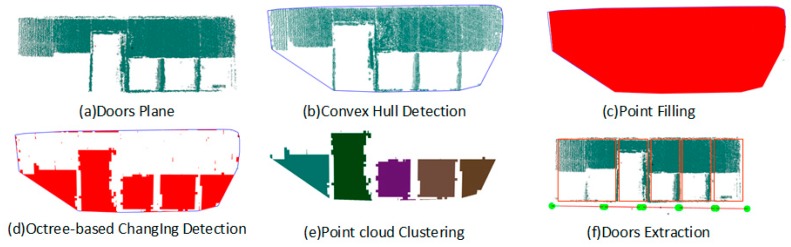
Generation of the opening components.

**Figure 9 sensors-19-00533-f009:**
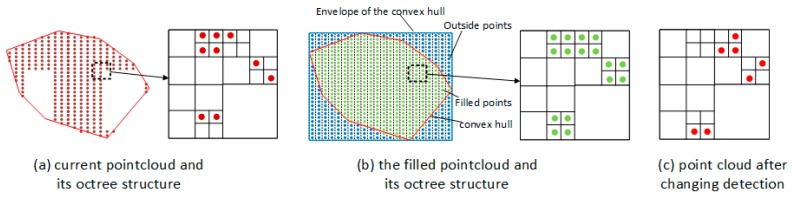
Point filling and changing detection methods.

**Figure 10 sensors-19-00533-f010:**
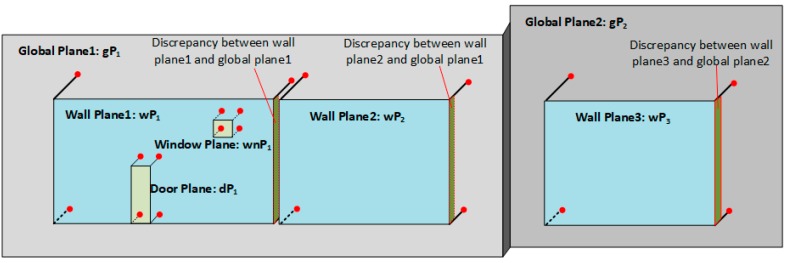
Strategy of global optimization for planes.

**Figure 11 sensors-19-00533-f011:**
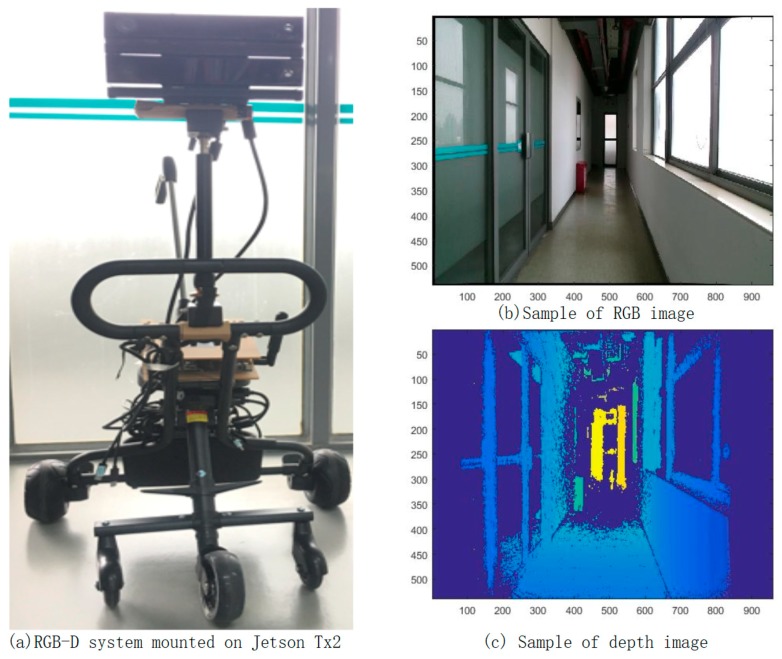
RGB-D system of data collection.

**Figure 12 sensors-19-00533-f012:**
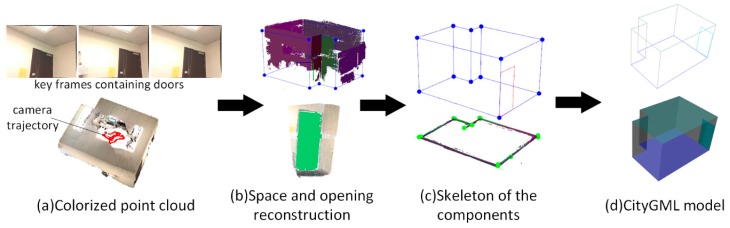
Reconstruction results for case study (1).

**Figure 13 sensors-19-00533-f013:**
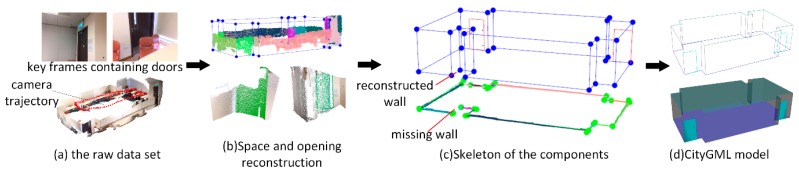
Reconstruction results for case study (2).

**Figure 14 sensors-19-00533-f014:**
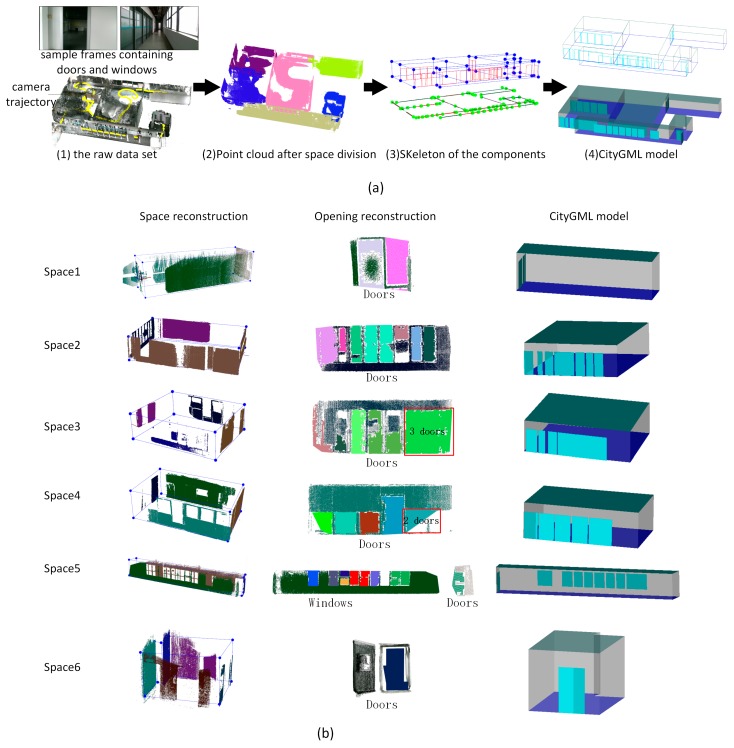
Reconstruction results for case study (3), (**a**) reconstruction results in different stages during indoor reconstruction of the whole model, (**b**) reconstruction results of each functional space in case study (3).

**Figure 15 sensors-19-00533-f015:**
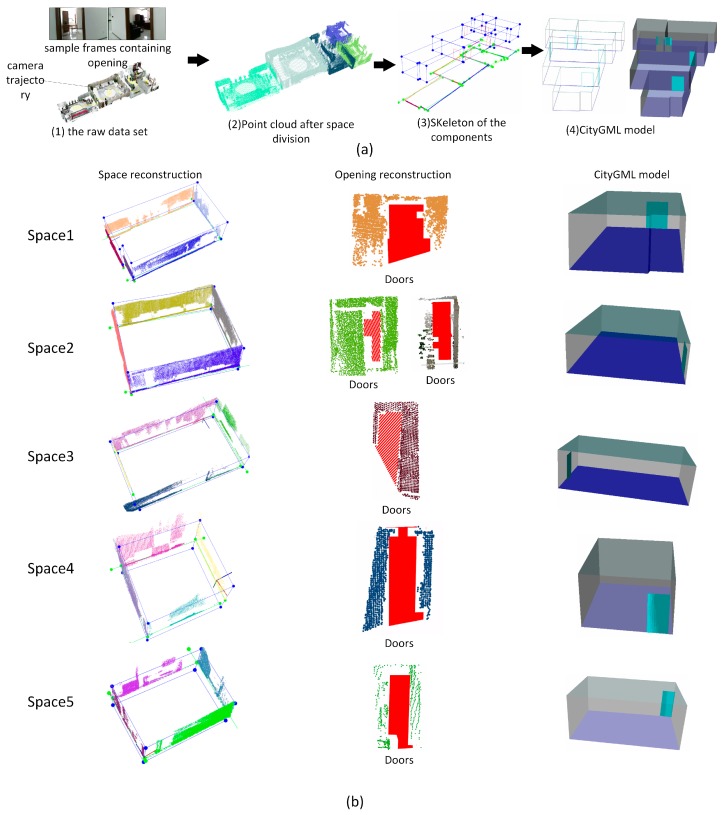
Reconstruction results for case study (4), (**a**) reconstruction results in different stages during indoor reconstruction of the whole model, (**b**) reconstruction results of each functional space in case study (4).

**Figure 16 sensors-19-00533-f016:**
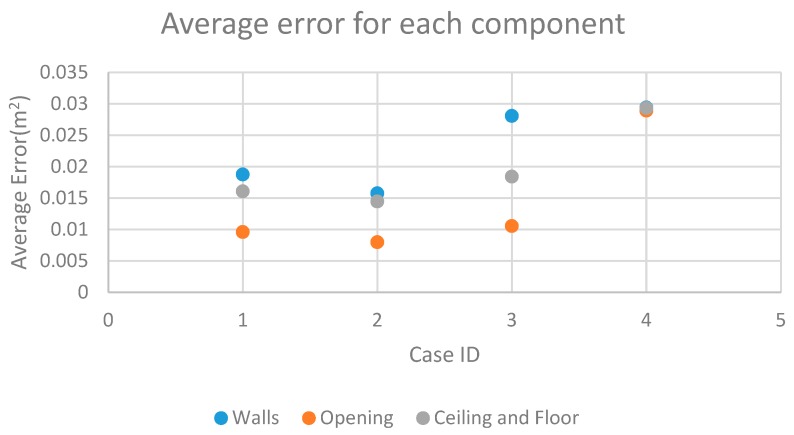
Average error for each component.

**Table 1 sensors-19-00533-t001:** Evaluation of the extracted components.

Case ID	Components	Recognized Number (RgN)	Recall Number (RcN)	Total Number (TN)	Accuracy (%)
1	Walls	6	0	6	100
	Door	1	0	1	100
	Ceiling	1	0	1	100
	Floor	1	0	1	100
2	Wall	14	1	14	100
	Door	2	0	2	100
	Ceiling	1	0	1	100
	Floor	1	0	1	100
3	Wall	26	1	29	89
	Door	23	0	25	92
	Window	8	0	9	88
	Ceiling	6	0	6	100
	Floor	6	0	6	100
4	Wall	22	1	23	95
	Door	6	0	6	100
	Ceiling	5	0	5	100
	Floor	5	0	5	100
**(Accuracy = RgN)/TN)**					

**Table 2 sensors-19-00533-t002:** Time consumption and the measurement accuracy of reconstructed components.

Case ID	Components	Time (s)	Recognized Area Dimension (m^2^)	Measured Area Dimension (m^2^)	Accuracy (m^2^)	Accuracy (%)	Average Accuracy (%)
1	Walls	23.2	42.72	41.93	0.79	98.13	97.23
	Door		2.03	2.01	0.019	99.04	
	Ceiling		14.28	14.06	0.22	98.39	
	Floor		14.28	14.06	0.22	98.39	
2	Walls	31.8	101.33	99.76	1.57	98.43	98.68
	Door		4.22	4.19	0.03	99.20	
	Ceiling		67.5	66.54	0.96	98.55	
	Floor		67.5	66.54	0.96	98.55	
3	Walls	84.8	468.14	455.36	12.78	97.19	98.21
	Door		46.04	45.56	0.48	98.95	
	Windows		15.56	15.35	0.21	98.63	
	Ceiling		266.16	261.35	4.81	98.16	
	Floor		266.16	261.35	4.81	98.16	
4	Walls	78.3	256.78	249.44	7.34	97.02	97.06
	Door		12.09	12.45	0.36	97.11	
	Ceiling		145.97	141.82	4.15	97.07	
	Floor		145.97	141.82	4.15	97.07	

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
