# Peer review of "Fast and Automatic Reconstruction of Semantically Rich 3D Indoor Maps from Low-quality RGB-D Sequences"

_sensors, 2019, doi:10.3390/s19030533_

Round 1

Reviewer 1 Report

This is a very interesting and well done paper.  I do have a couple of concerns / recommendations. I found Section 3.3.2. confusing.  You refer to the curvature of a point.  How can a point have curvature? Please explain. Also you refer to the normal to a point. A point on its own cannot have a normal, please expand your presentation to clarify this. Also you say {Lines 253-254} "the seeds container iteratively grows until is is empty" - please rephrase this so it makes sense.

Section 3.4, particularly the paragraphs {Lines 338 - 359} you refer to a "hull polygon as a red solid line, ... points outside as blue dots", neither of which I can see.  Also your description of the changing detection method is not clear, e.g., it appears you are inverting the occupancy of the voxels in the octree but this in not clear explained in the text. Please review this section closed and work to make it more understandable. 

Author Response

First of all, we would like to express our sincere appreciation to the comments and suggestions from the reviewers!

Reviewer 2 Report

In general topic is interesting and up-to date. The Algorithm works quite correctly.

Description is not clear at a few points, described below.

General remarks:

Authors claim the method faster and more robust, while there is no comparison with other methods (accuracy, time of processing).

Authors suggested automatic 3D reconstruction, while sequence was manually tagged by room names. It has to be clarified in text.

It was not described how the test sequences were obtained and tagged with camera positions. Only mentioned the "low quality".

The example no 3 with a few rooms seems to be very specific, evidently not a standard multi room space. For showing the robustness of the method, more common example should be selected.

There is no information about time of computation.

Other remarks below in detailed part.

Closely related recent work (not cited):

Jianwei, et all, Robust and Efficient CPU-Based RGB-D Scene Reconstruction Sensors 2018, Vol. 18 (11)

Detailed remarks:

page3 line117: “[28] presented an automated method to create accurate 2D floor plane models of building interiors from a 3D point cloud [28]”

Double citation. First should be replaced by name of author.

p4 l156: “In this study, we seek to establish an automatic indoor reconstruction method”. Method is semi-automatic? Requires manually marked doors (called start-stop point)

p5 l180: “Because RGB-D frames are labeled by camera position and orientation, timestamp, and semantic tags”

What is a source of so rich RGB sequence? While camera position can by automatically obtained by SLAM method, who tagged sequence with semantic tags? What are semantic tags in this case?

p7. Fig. 4.: How the camera trajectory was divided before recognition of the doors? It is a weak point of the algorithm. It should be described how the doors (division of the spaces) are recognized. Specifically, how to distinct two spaces from single space with divided by half-wall?

Everything later is based on this division, because subsequent steps assume processing of single room.

p8 l253: The seeds container iteratively grows until it is empty. ??? Until what is empty? Seed container? Probably “until there is no new candidate point”

p8 l257: (d) 2D Porjection

p8 l264: The maximum of the plane length is larger ... What is it the length of plane?

p8 l274: “obtain the optimal fitted line ?|{? = ?? + ?}.” “The fitted line consists of two endpoints”

The word “line” is used sometimes as geometrical line (infinite) and sometimes as segment of line (with two endpoints). It leads to misunderstandings and should be clarified.

p11 l306: “Meanwhile, the corresponding intersection points are calculated“ Are the lines a geometrical lines (extrapolated)? Or line segments with endpoints? If segments - are the segments extrapolated?

p11 l307:  “It continues until the algorithm returns back to the starting line.”  What if it return back to the starting line and some lines are still not processed? It can happen if space division is incorrects (p7. fig. 4)

p13 fig.8.: It is very unusual example of the door. It;s hard to see the connection between scan (Fig. 8a) and recognized doors (Fig. 8f).

p13 l332: “Because we interactively define the start-end positions for each functional space” Is it done manually?

p13 l338: “First, a convex hull algorithm is used to compute the envelope of the projected planes” Projected on what?

p13 l344: A few wrong fig numbers. “Figure 8(b) shows the convex hull polygon as a red solid line” Ref to fig. 9

p16 tab.1.: “(Accuracy=(RgN+RcN-RFN)/TN)” What is RFN? Isn’t it RgN/TN?

Accuracy with 4 digit number (89.66% )is pointless. 89% is enough.

In tables 1 there is an accuracy but in table 2 there is an error. It is inconsistent. Would be better to use one of them.

p17 l459: “Table 2lists the area dimension of the recognized components and the area dimension of measured components for comparison purposes.”

Area dimension?? -> surface area

“measured” means “actual”?

p17 l464: “dimension” means area?

p18 l466: “We can conclude that smaller area dimensions tend to result in a 467 higher accuracy”.

This conclusion is not supported by data.

For example error on walls is averaged, and to get such conclusion, in fig. 13 should be average size of wall, floor etc.

p19 fig. 14.: For space 4, one of the scan in column 1 and 2 seems to be upside down.

p20 l493: “This provides a faster, more convenient, and lower-cost solution for semantically rich 3D indoor mapping”.  Faster than what? There is not data about time of calculation of presented algorithm  nor comparison to other method.

p 20 l493: “enables the full use of information provided by the RGB-D mapping system” Not true, RGB data not used at all.

Author Response

(The authors gave the same response as above.)

Round 2

Reviewer 2 Report

Corrections accepted.